# AI-Based Detection of Unwanted Behavior: The Paradoxical Effect of Standard Data Augmentation in Video Surveillance

**Maksim V. Emelianov**
Department of Mechanics and Mathematics
Novosibirsk State University (NSU)
Novosibirsk, Russia
m.emelyanov1@g.nsu.ru

## Abstract

Public spaces and commercial environments face persistent challenges regarding human misconduct. Traditional surveillance remains passive, while manual monitoring is labor-intensive and inefficient. Consequently, automating the detection of unwanted behavior via digital assistants is essential. This study explores the application of deep learning models to identify such behavior and analyzes the impact of data augmentation on model performance.

We utilized the UCF-Crime dataset (1,900 videos, 13 classes) and constructed a novel custom dataset comprising 5,236 videos with a focus on "violent behavior." Algorithms based on ResNet3D and DenseNet-RNN were trained on both original and augmented versions of these datasets. On the UCF-Crime dataset, the DenseNet-RNN model showed stability only in detecting the "arson" class. However, the model's stability and recognition quality improved significantly when trained on the custom dataset.

Crucially, our experiments demonstrated that data augmentation negatively impacted the recognition quality on the custom dataset ($F(1, 36) = 7.40, p = 0.01$). Specifically, the ResNet3D method exhibited a significant degradation in performance, with the AUC dropping from 0.83 (95% CI: $[0.80 - 0.86]$) before augmentation to 0.73 (95% CI: $[0.69 - 0.77]$) post-augmentation. The DenseNet-RNN method showed a similar downward trend (AUC 0.82, 95% CI: $[0.79 - 0.85]$ vs. AUC 0.75, 95% CI: $[0.71 - 0.79]$). These findings suggest that the blind application of standard augmentation procedures—a common industry practice for both static images and video frames—can lead to a paradoxical decline in detection quality. This highlights the urgent need to critically reevaluate how augmentations are selected and to develop task-specific methodologies that ensure transformations actually improve, rather than hinder, model efficacy in behavioral recognition.

## 1 Introduction

In modern society, public spaces and commercial environments frequently face challenges related to visitor misconduct. These unwanted behaviors range from minor rule violations to severe criminal activities, such as vandalism, assault, and physical violence (Ansari & Singh, 2022). Such incidents inevitably lead to property damage and significant safety risks, making the automated monitoring of mass-gathering areas a critical priority for public security.

To mitigate these risks, security stakeholders heavily rely on Closed-Circuit Television (CCTV) systems to monitor human activity. Despite their widespread deployment, traditional surveillance systems remain inherently passive. Due to the limitations of human attention and the cognitive fatigue associated with monitoring numerous screens simultaneously, manual surveillance is increasingly labor-intensive and economically inefficient. Consequently, CCTV is often relegated to a forensic tool used only after an incident has occurred, thereby diminishing the return on costly security investments.

To enhance operational efficiency, there is a critical need for intelligent video surveillance systems capable of autonomously detecting unwanted behaviors in real-time. Recent advancements in deep learning, particularly Convolutional Neural Networks (CNNs) and Recurrent Neural Networks (RNNs), have shown significant promise in video processing and action recognition tasks. However, predicting human behavior in complex environments remains a formidable challenge. The performance of these intelligent systems heavily relies on the quality of the training data and the chosen network architectures. Existing datasets often suffer from low resolution, extreme class imbalance, and a lack of precise spatial-temporal annotations, which collectively impede the generalization capabilities of predictive models.

In this study, we propose a neural network-based framework for the detection of unwanted human behavior in general surveillance settings. Our contributions are threefold:

- We address the limitations of existing public datasets (such as UCF-Crime) by constructing and distilling a novel custom dataset of 5,236 video clips, heavily annotated and focused specifically on violent behavior.
- We adapt and evaluate two distinct architectures—a hybrid DenseNet-RNN model and a ResNet3D model—for the task of behavioral screening in constrained real-world computing environments.
- We systematically investigate the impact of online data augmentation on model performance. Surprisingly, our results reveal a paradoxical effect: standard augmentation techniques significantly degrade the detection capabilities of the models in this specific domain.

The remainder of this paper is structured as follows: Section 2 reviews existing literature and datasets. Section 3 details our proposed methodology, including dataset preparation and network architectures. Section 4 presents our experimental results, followed by a discussion of the augmentation paradox in Section 5. Finally, Section 6 concludes the paper.

## 2 Related Work

**Video Anomaly Detection (VAD) and Datasets.** The detection of anomalous or unwanted behavior in video streams has been extensively studied (Sernani et al., 2021). The efficacy of predictive models heavily depends on the diversity and quality of the training datasets. Several public datasets have been introduced to address this, including the Avenue Dataset (Lu et al., 2013) and the DCSASS dataset. Notably, the UCF-Crime dataset (Sultani et al., 2018) provides a large-scale collection of 1,900 untrimmed surveillance videos covering 13 anomaly classes (e.g., arson, assault, shoplifting). While comprehensive, UCF-Crime and similar datasets often contain excessive background noise, lack precise temporal boundary annotations, and suffer from severe class imbalances, making fine-grained behavior detection challenging.

**Action Recognition Architectures.** Traditional approaches to video analysis have transitioned from hand-crafted features to deep learning architectures. Standard 2D Convolutional Neural Networks (CNNs) excel at spatial feature extraction but fail to capture temporal dynamics. To address this, hybrid models combining CNNs with Recurrent Neural Networks (e.g., LSTMs) have become prominent for sequential data modeling. More recently, 3D Convolutional Networks (like ResNet3D) have demonstrated superior performance by simultaneously learning spatial and temporal representations directly from video volumes. While recent state-of-the-art methods (such as Video Transformers) achieve exceptional accuracy on public benchmarks, they often incur prohibitive computational costs. In contrast, traditional architectures like DenseNet (Huang et al., 2017) and ResNet (He et al., 2016) remain highly effective and are significantly less computationally demanding, which is an absolute necessity for deployment on resource-constrained edge devices in real-world surveillance systems.

## 3 Methodology

Our approach to detecting unwanted behavior involves evaluating two computationally efficient deep learning architectures under various data conditions. We specifically focus on the impact of data augmentation and the refinement of training datasets.

## 3.1 DATASETS AND DATA DISTILLATION

Initially, we utilized the publicly available **UCF-Crime Dataset**, which encompasses 128 hours of footage across 13 classes of anomalous behavior. However, preliminary experiments revealed that the presence of prolonged normal segments within these videos diluted the learning signal. The lack of explicit spatial-temporal localization for the individuals exhibiting the unwanted behavior hindered the network's ability to extract relevant features.

To overcome these limitations, we constructed a **Custom Distilled Dataset**. This dataset specifically targets a single, well-defined class of unwanted behavior: "violent behavior" (encompassing fights and physical aggression). It consists of 5,236 curated video clips ranging from 5 to 60 seconds. The data underwent a rigorous "distillation" process, incorporating precise temporal trimming to isolate the anomaly and spatial tracking via YOLO to focus the region of interest (ROI) on the acting individuals. Importantly, YOLO proved highly robust to motion blur during fast physical strikes, retaining human silhouettes effectively. Natural occurrences of individuals briefly leaving the frame or being occluded were preserved in the dataset, ensuring that the tracking process did not introduce artificial selection bias (e.g., dropping hard-to-detect violent frames). This dataset provides a highly balanced, noise-reduced environment for training.

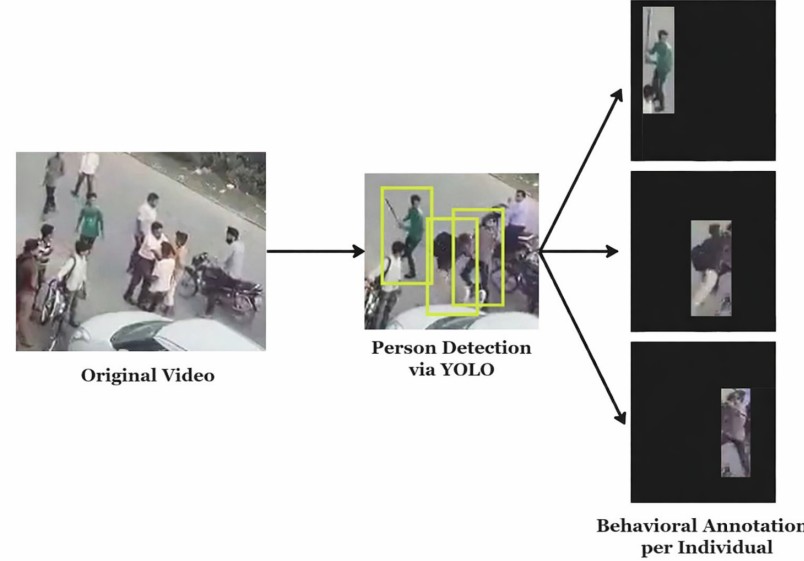

Figure 1: Spatial-temporal annotation and distillation process of the custom dataset, highlighting the region of interest (ROI) and temporal boundaries of violent behavior.

## 3.2 NETWORK ARCHITECTURES

We deployed two distinct architectures optimized for video sequence analysis:

**1. DenseNet-RNN Hybrid Model:** This architecture processes videos sequentially. Each video is divided into frames, and a pre-trained 121-layer DenseNet operates as a spatial feature extractor, benefiting from dense connectivity to maximize feature reuse. The resulting 1024-dimensional feature vectors from sequences of four consecutive frames are fed into a 2-layer Long Short-Term Memory (LSTM) network (Graves, 2012) (hidden state dimension of 512). The LSTM models the temporal evolution of the features, outputting a context vector that is passed through a fully connected layer for final classification.

**2. ResNet3D:** To inherently capture spatiotemporal features, we utilized a modified ResNet-2p1d architecture with a depth of 152 layers. Unlike the 2D convolutions in standard ResNets, the 3D convolutional kernels operate across spatial dimensions and the temporal time-step. We optimized the input window to process sequences of four frames, leveraging residual connections to facilitate deep feature extraction without the vanishing gradient problem.

### 3.3 Training Strategy and Online Augmentation

Both models were trained using a standard split strategy with Cross-Entropy Loss (or Binary Cross-Entropy with Logits Loss for the custom dataset) and the Adam optimizer (Kingma & Ba, 2014). To prevent overfitting, we applied $L_2$ regularization.

A critical aspect of our methodology was the application of **Online Data Augmentation**. To artificially expand the training distribution, we applied a stochastic pipeline during the training phase. Transformations included:

- Severe spatial transformations ($180°$ rotation) and subtle geometric shifts (random tilt/rotation, horizontal flipping).
- Random scaling ($0.8 - 1.2$) and cropping ($100 - 200$ pixels).
- Photometric distortions including additive random noise, brightness adjustments, and grayscale conversion.

This specific bundle of transformations was deliberately chosen because it represents a standard, widely adopted pipeline in modern image recognition workflows, where it is often applied blindly to increase data volume. While these techniques are ubiquitous in computer vision to improve generalization, our subsequent experiments critically evaluate their efficacy in the specific context of automated behavioral screening.

## 4 Experiments and Results

To rigorously evaluate the proposed architectures and understand the impact of data augmentation, we conducted a series of cross-validation experiments. Model performance was primarily assessed using the Area Under the Receiver Operating Characteristic Curve (AUC-ROC) to provide a threshold-invariant metric. We report the AUC alongside its 95% Confidence Interval (CI). Additionally, we captured momentary metrics—Accuracy, Precision, Recall, F1-Score, and Specificity—to provide a comprehensive performance profile.

### 4.1 Baseline Evaluation on the UCF-Crime Dataset

Initial experiments were conducted using the multi-class, unaugmented UCF-Crime dataset to establish a baseline for the DenseNet-RNN hybrid model. The goal was to classify 13 distinct types of anomalous behavior alongside normal surveillance footage.

The results indicated severe instability and generally poor predictive performance across most classes, yielding an average AUC of approximately 0.51. The model struggled significantly with minority classes; for instance, "Assault" and "Vandalism" demonstrated highly unstable AUC trajectories during cross-validation, occasionally dropping below 0.50, effectively indicating inverted or random predictions. Notably, out of the 13 categories, the model exhibited stability only in detecting the "Arson" class (see Figure 2). We hypothesize that the distinct, persistent visual signature of fire allowed the spatial feature extractor to consistently identify this anomaly, despite the background noise.

Overall, this exploratory evaluation on UCF-Crime confirmed that without explicit spatial-temporal bounding, traditional architectures struggle to learn from long, untrimmed videos dominated by background frames. Rather than serving as a direct benchmark comparison, this initial failure necessitated the shift to our Custom Distilled Dataset, changing the task to a focused, binary behavior classification problem.

### 4.2 Performance on the Custom Distilled Dataset

By transitioning to the Custom Distilled Dataset—which isolated the "violent behavior" class using precise spatial and temporal tracking—we observed a marked improvement in model stability and absolute performance metrics, independent of the applied architecture (as illustrated by the tight clustering of black squares in Figure 2).

When trained without data augmentation, the **ResNet3D** architecture achieved robust classification results. The model recorded an AUC of 0.83 (95% CI: $[0.80 - 0.86]$). The momentary metrics further validated its efficacy as a screening tool, achieving an Accuracy of 80.3%, Precision of 65.0%, Recall of 65.8%, and an F1-Score of 65.4%.

Similarly, the **DenseNet-RNN** model without augmentation demonstrated comparable baseline efficiency, reaching an AUC of 0.82 (95% CI: $[0.79 - 0.85]$). This indicates that both spatial-temporal sequence modeling and 3D convolutions are highly viable approaches for this domain when trained on refined data.

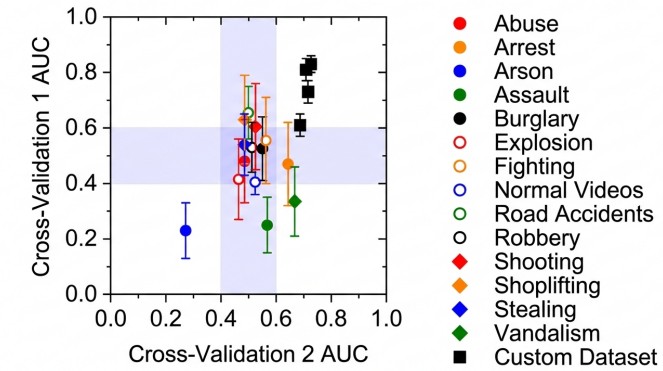

Figure 2: Cross-validation AUC stability comparison. Scatter plot shows the relationship between two cross-validation iterations. Classes from the UCF-Crime dataset display wide dispersion (low stability), whereas the models trained on the Custom Distilled Dataset (represented by black squares) exhibit tight clustering in the upper-right quadrant, indicating both high accuracy and robust stability regardless of the specific architecture.

### 4.3 THE AUGMENTATION PARADOX: STATISTICAL ANALYSIS

In standard ML workflows, data augmentation is universally applied under the assumption that artificially expanding the training distribution inherently improves model generalization. To test this assumption in the context of behavioral screening, we subjected the Custom Distilled Dataset to the aggressive stochastic augmentation pipeline detailed in Section 3.

Paradoxically, the application of this standard bundle resulted in a statistically significant degradation of detection quality. A repeated-measures analysis of variance revealed a strong negative main effect of augmentation on the models' predictive capabilities ($F(1, 36) = 7.40, p = 0.01$).

Table 1: Impact of Data Augmentation on Model Performance (Custom Dataset)

| Architecture | Condition | AUC | 95% CI | Accuracy |
|---|---|---|---|---|
| ResNet3D | No Augmentation | **0.83** | $[0.80 - 0.86]$ | **0.803** |
| ResNet3D | With Online Augmentation | 0.73 | $[0.69 - 0.77]$ | 0.666 |
| DenseNet-RNN | No Augmentation | 0.82 | $[0.79 - 0.85]$ | 0.762 |
| DenseNet-RNN | With Online Augmentation | 0.75 | $[0.71 - 0.79]$ | 0.697 |

As summarized in Table 1, the **ResNet3D** method experienced a steep decline in performance. The AUC dropped from 0.83 down to 0.73, and the confidence interval shifted completely downwards ($[0.69 - 0.77]$ post-augmentation). This performance degradation was statistically significant ($p = 0.02$). Furthermore, the momentary metrics for ResNet3D collapsed under augmentation: Accuracy fell to 66.6%, and Precision dropped severely to 49.7%, generating an unacceptable rate of false positives for a surveillance system.

The **DenseNet-RNN** architecture exhibited a parallel downward trend. The baseline AUC of 0.82 degraded to 0.75 under augmented conditions. While the isolated statistical drop for DenseNet-RNN

was slightly less pronounced ($p = 0.17$), the consistent trajectory across entirely different network paradigms confirms the overarching negative impact.

These findings robustly demonstrate that standard spatial and pixel-level augmentations—rather than aiding the neural networks—corrupted the delicate spatial-temporal signatures required to identify human violence.

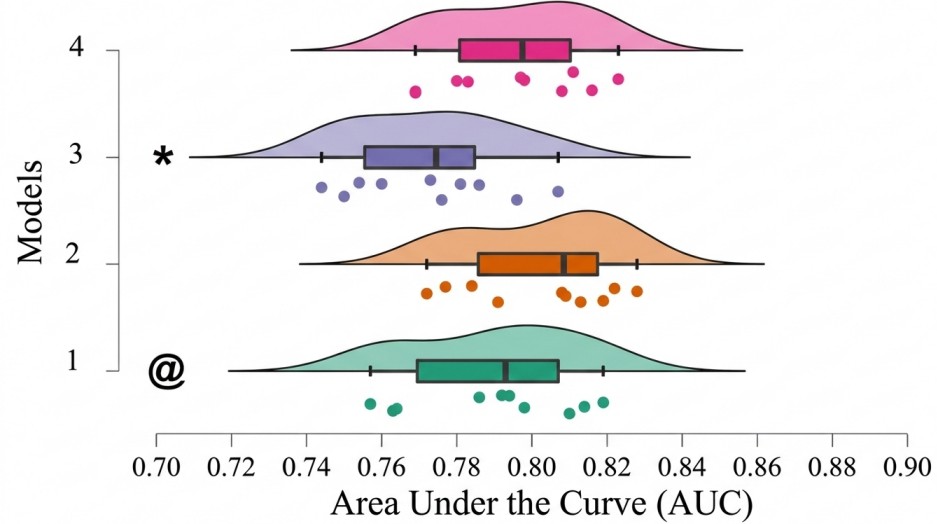

Figure 3: Density and boxplot distributions of AUC scores across different modeling approaches on the Custom Distilled Dataset. Models without data augmentation (Model 4: ResNet3D, Model 2: DenseNet-RNN) consistently outperform their counterparts trained with online data augmentation (Model 3: ResNet3D, Model 1: DenseNet-RNN). The application of augmentation causes a visible leftward shift, demonstrating the augmentation paradox. Symbols denote statistical comparisons against the respective unaugmented baselines: $*$ indicates $p = 0.02$ (Model 3 vs. Model 4), and @ indicates $p = 0.17$ (Model 1 vs. Model 2). The shaded curves represent the probability density of AUC values, while the embedded boxplots display the median, interquartile range (25th and 75th percentiles), and non-outlier boundaries.

## 5  DISCUSSION

The objective of this study was to evaluate lightweight, robust architectures (DenseNet-RNN and ResNet3D) for the automated detection of unwanted behavior in general surveillance environments. Initially, we hypothesized that the combination of spatial and recurrent layers would yield high AUC scores ($>0.80$) on established public datasets like UCF-Crime. However, empirical results contradicted this expectation, yielding an average AUC of 0.51 across 13 classes. Detailed class-wise analysis revealed that only the "Arson" class was reliably detected. We attribute this exception to the persistent, high-contrast, and relatively static visual signature of fire, which the spatial feature extractor could easily isolate. Conversely, behaviors defined by complex, subtle human interactions (e.g., "Assault" or "Shoplifting") were lost in the unannotated, lengthy background footage of the UCF-Crime dataset.

This limitation justified the creation of our Custom Distilled Dataset, which focused exclusively on violent behavior with strict temporal and spatial bounding. As expected, baseline models trained on this distilled dataset demonstrated a substantial leap in both absolute performance and cross-validation stability.

**Explaining the Augmentation Paradox.** The most significant finding of this study is the paradoxical degradation of model performance upon the application of standard online data augmentation. While techniques such as geometric transformations and color-space shifts are universally

recommended to prevent overfitting in static object recognition, they actively corrupted the learning process in our behavioral screening task.

We propose several mechanistic explanations for this phenomenon:

- **Disruption of Biomechanical Semantics:** Human violence involves specific physical postures and gravity-dependent dynamics (e.g., falling, striking). Severe geometric transformations, such as $180°$ rotations or extreme horizontal flipping, distort the natural physical constraints of these actions. The network is forced to learn physiologically impossible movements, thereby diluting the discriminative features of genuine violence.

- **Loss of High-Frequency Interaction Details:** Additive noise and low-resolution scaling obscure critical, high-frequency details—such as the rapid intersection of limbs or facial expressions—that differentiate an aggressive physical altercation from an innocuous interaction (e.g., a hug or a fast-paced game).

- **Color and Texture Distortion:** Grayscale conversion and aggressive brightness shifting destroy subtle physiological and environmental cues. For instance, the contrast between the clothing of two distinct individuals is a crucial feature for the spatial extractor to separate overlapping human bounds during a fight. Removing color information forces the network to rely solely on edges, which are often noisy in CCTV footage.

While it may seem intuitive in hindsight that severe geometric transformations (like $180°$ rotations) destroy biomechanical semantics, such transformations are routinely included in generic augmentation libraries and applied blindly across the industry to artificially boost dataset sizes. The paradox lies in the fact that a practice universally assumed to improve generalization actively harms it in this context. These findings emphasize that the blind application of standard augmentation bundles is fundamentally flawed for spatiotemporal data. We acknowledge that grouping highly destructive transformations with subtle photometric shifts into a single stochastic pipeline limits our ability to isolate the precise cause of the degradation. Conducting a detailed ablation study to isolate the independent effects of each transformation (e.g., random cropping vs. rotation) is the immediate next step in our future work. Ultimately, it is imperative to move away from arbitrary augmentation pipelines and establish rigorous, task-specific methodologies that verifiably enhance model efficiency in behavioral recognition.

## 6 CONCLUSION

In this paper, we addressed the challenge of detecting unwanted human behavior in video surveillance under constrained computational resources. Our contributions and findings are summarized as follows:

1. We demonstrated that public, untrimmed datasets (like UCF-Crime) are insufficient for fine-grained behavioral detection using standard sequence models, as they only reliably expose anomalies with distinct static signatures (e.g., arson).

2. To solve this, we curated and distilled a novel custom dataset comprising 5,236 annotated video clips focused on violent behavior, which drastically improved the baseline stability of both ResNet3D and DenseNet-RNN architectures.

3. We uncovered a significant *augmentation paradox*: the application of standard online data augmentation statistically degraded model performance ($p < 0.05$), lowering the AUC and destroying precision.

These results emphasize that compact neural network architectures are highly capable of screening unwanted behavior, provided the training data is carefully distilled. Future research should focus on developing domain-aware augmentation strategies specifically tailored for human kinematics and interaction dynamics, avoiding the destructive pitfalls of standard image-level transformations. Ultimately, these insights lay the groundwork for deploying modular, highly accurate AI assistants in general security systems, advocating for a paradigm shift from blind data augmentation to methodologically justified, task-specific data transformations.

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
