# OpenReview forum: "AI-Based Detection of Unwanted Behavior: The Paradoxical Effect of Standard Data Augmentation in Video Surveillance"
_mathai.club/MathAI/2026/Conference — 2026 Oral_

### Official Review · Reviewer_UPa4 · 2026-03-11
**Review of 'AI-Based Detection of Unwanted Behavior: The Paradoxical Effect of Standard Data Augmentation in Video Surveillance'**

**Rating:** 5
**Confidence:** 4

**Review:**

The paper studies automated detection of unwanted behavior in surveillance videos using deep learning models. Two architectures, DenseNet-RNN and ResNet3D, are evaluated on the public UCF-Crime dataset and on a custom curated dataset focused on violent behavior. The work primarily investigates the impact of standard data augmentation techniques on model performance. The experiments suggest that commonly used augmentation strategies (e.g., rotation, noise, brightness shifts, cropping) can degrade detection performance in this context, highlighting the need for more task-specific augmentation strategies.

Strengths

1. Addresses an important practical problem in automated surveillance, namely the reliable detection of unwanted or violent behavior in video streams.
2. Provides an empirical investigation into the impact of standard data augmentation techniques on behavioral recognition models.
3. Evaluates two different architectures (DenseNet-RNN and ResNet3D), offering some comparison between sequence-based and 3D convolutional approaches.
4. Highlights a potentially interesting observation that generic image-based augmentations may not always transfer well to spatiotemporal behavioral recognition tasks.

Weaknesses/Limitations

1. Limited methodological novelty: The work mainly evaluates existing architectures and training strategies rather than proposing a new model or algorithm. However, this is understandable since the primary contribution of the paper is empirical.
2. Limited detail about the custom dataset: The paper introduces a distilled dataset of 5,236 clips focused on violent behavior, but the dataset construction and annotation process are only briefly described. Although page limits may have restricted a more detailed explanation, providing additional details on data preparation and dataset splits (either in the main text or by adding an appendix) would improve clarity and reproducibility.
3. Limited experimental scope: The experiments focus on two architectures and an augmentation pipeline. While the current methodology is reasonably structured, evaluating additional augmentation strategies or performing a small ablation study on individual transformations could provide a clearer understanding of which augmentations contribute most to the observed effect.
4. Potential dataset-specific conclusions: Most results are reported on the custom distilled dataset focused on a single behavior class. Additional validation on other datasets could help assess the generality of the findings, although the current results still provide useful insights within the studied setting.

Overall, the paper explores an interesting empirical observation regarding the impact of standard augmentation techniques in behavioral video analysis. While the results may provide useful practical insights, the methodological novelty and experimental depth are somewhat limited. Providing more details about the dataset and expanding the experimental evaluation could further strengthen the contribution.

---

> ### Author Rebuttal · Authors · 2026-03-13
>
> We sincerely thank Reviewer for their thoughtful evaluation and constructive feedback. We are highly encouraged that the reviewer recognizes the practical importance of our work in automated surveillance and the value of our empirical investigation. While our core contribution is empirical rather than architectural, statistically exposing the paradoxical degradation caused by standard image-based augmentations in spatiotemporal models represents a critical conceptual shift for the Video Anomaly Detection community. By highlighting this anomaly, our work challenges a widely accepted industry default and underscores the urgent need to critically reevaluate data preparation pipelines in behavioral recognition.
>
> We completely agree with the reviewer's valid observation regarding the limited details provided about our Custom Distilled Dataset. The description of the dataset construction was unintentionally over-condensed during the adaptation of this manuscript from a broader, more comprehensive underlying research project. We fully acknowledge that thorough documentation is vital for reproducibility. To resolve this in the camera-ready version, we will expand the methodology section and include a comprehensive Appendix dedicated entirely to the dataset. This supplementary material will explicitly describe the video sourcing process, the manual spatial-temporal annotation and trimming methodology used for distillation, as well as the exact train, validation, and test splits utilized during our cross-validation experiments.
>
> Regarding the excellent suggestions to perform an ablation study on individual transformations and to validate the findings across other datasets, we wish to clarify the intended scope of this paper. Our primary objective in this initial study was to robustly identify and statistically confirm the existence of this overarching augmentation anomaly. Determining exactly which specific types of augmentation—such as geometric distortions versus photometric shifts—drive the worst degradation is the immediate next step in our ongoing research. In the revised manuscript, we will significantly expand our Discussion section to explicitly frame this limitation. We will outline the necessity of granular ablation studies and broader validation across multiple behavioral datasets as critical directions for future work, setting a clear roadmap for developing methodologically justified, task-specific augmentation strategies.

---

### Official Review · Reviewer_u9d1 · 2026-03-13
**The authors construct a custom, temporally-cropped dataset for video anomaly detection (VAD) and evaluate two deep learning architectures, claiming to discover an "augmentation paradox" wherein standard data augmentations significantly degrade model performance.**

**Rating:** 4
**Confidence:** 4

**Review:**

Strengths:
The problem of automated video anomaly detection in resource-constrained environments is undeniably important.
The manual curation and bounding of 5,236 video clips is a commendable engineering effort.

Weaknesses:
The authors apply spatial rotations and horizontal flipping to human action recognition datasets and show that this results in performance drop. However, I believe, proposed augmentations are not "standard industry practice". Gravity is directional; flipping CCTV footage upside down destroys biomechanical semantics.
The authors rely exclusively on two architectures (DenseNet-RNN and ResNet-3D). Comparisons against modern models are noticeably absent.
The authors assert that "standard augmentation" is harmful, yet they don't show any ablation studies. The authors grouped highly destructive transformations (180° rotation) with standard ones (cropping, subtle brightness shifts) into a single stochastic pipeline. Without isolating which augmentation caused the performance drop - the core claim is unjustified.

Questions:
It will be interesting to see ablation study isolating the effect of each individual augmentation technique (e.g., what is the isolated effect of random cropping vs. rotation?).
Consider to replicate your experiments using at least one SOTA video transformer to prove that paradox is not an artifact of CNN/RNN inductive biases.
What is the false-negative rate of the YOLO object detector used to distill your dataset? If YOLO fails on blurry, violent frames, those frames are dropped, introducing a massive, unmeasured selection bias into your training distribution. Please quantify this.

---

### Official Review · Reviewer_BMPP · 2026-03-13
**The empirical observations are potentially interesting, but the paper does not establish a strong research contribution**

**Rating:** 4
**Confidence:** 4

**Review:**

1) Summary

This paper addresses unwanted-behavior detection in surveillance video, with the final focus narrowed to violent behavior. The authors position the work around three contributions: a custom distilled dataset of 5,236 annotated violent-behavior clips, an empirical comparison of two lightweight architectures for this setting (DenseNet-RNN and ResNet3D), and the claim that standard online augmentation is harmful rather than helpful for behavioral video recognition. Experimentally, the paper first evaluates DenseNet-RNN on the original 13-class UCF-Crime benchmark and reports weak performance, with average AUC around 0.51 and stability only for the arson category. It then moves to the distilled violent-behavior dataset, where the reported unaugmented results improve to AUC 0.81 for DenseNet-RNN and 0.83 for ResNet3D. The paper presents this shift as evidence that refined spatial-temporal annotation is more important than model choice in this domain, and that generic image-style augmentation can corrupt the cues needed for violence recognition

2) Strengths

The paper addresses a practically relevant problem in surveillance video analysis and presents the experimental setup in a clear and accessible way. The motivation for moving from long untrimmed anomaly videos toward a more tightly localized violent-behavior setting is explained coherently, and the comparison between DenseNet-RNN and ResNet3D provides a useful empirical perspective on two standard modeling choices for this task. The manuscript also gives appropriate attention to a negative experimental result: the observed performance decrease under the proposed augmentation pipeline is explicitly discussed and incorporated into the paper’s overall conclusions. More broadly, the study offers a readable and well-structured account of how data refinement, model choice, and augmentation interact in this application setting.

3) Weaknesses

The main weakness is limited scientific novelty. The architectures are standard, and the central empirical claim is also narrower than it is presented. The paper positions the augmentation paradox as its main finding, but the broader point that augmentation for video understanding must preserve action semantics is already known in prior work; for example, earlier video-action papers explicitly note that naïve horizontal flipping can be problematic, and later work motivates task-specific augmentation precisely because generic image-style transforms do not always preserve spatio-temporal semantics.

A second limitation is the lack of positioning against current strong baselines. The related work section mainly cites foundational datasets and generic backbone papers, but does not compare against stronger recent methods on public benchmarks. This is important because substantially stronger public results already exist. On violence-specific benchmarks, CUE-Net reports 94.0% accuracy on RWF-2000 and 99.5% on RLVS, while MSTFDet reports 95.2% on RWF-2000. On UCF-Crime, recent methods such as REWARD and GS-MoE report 86.94% and 91.58% AUC, respectively. Even allowing for differences in datasets and metrics, the paper does not establish a state-of-the-art contribution and does not clearly position its results relative to the current literature.

The dataset comparison is also less conclusive than the paper suggests. The manuscript does report the same architecture on both datasets for DenseNet-RNN, with average AUC increasing from about 0.51 on UCF-Crime to 0.81 on the custom dataset. However, this is not a controlled comparison of the same task under two data sources: the paper moves from 13-class anomaly classification on long untrimmed videos to a binary violent-behavior problem after temporal trimming and ROI tracking. Therefore, the improvement cannot be attributed solely to dataset distillation; it also reflects a substantial change in label space and task difficulty. In addition, ResNet3D is only reported on the custom dataset, not on UCF-Crime, so the cross-dataset story is incomplete at the model level.

The augmentation claim is also not sufficiently isolated experimentally. The training pipeline combines 180° rotation, horizontal flipping, noise, grayscale conversion, scaling, and cropping. Since several of these transforms can plausibly damage behavior-specific motion and interaction cues, the reported degradation is not fully surprising. In its current form, the paper shows that this particular augmentation bundle harms performance in this particular setting. Without a per-transformation ablation, it does not yet support a broader conclusion about augmentation for surveillance behavior recognition.

Finally, the quantitative reporting contains internal inconsistencies. The paper states that AUC is reported with 95% confidence intervals, yet Section 4.2 gives the ResNet3D result as a 90% CI. There are also mismatches between the abstract, the running text, and Table 1 for the post-augmentation confidence intervals: DenseNet-RNN is reported as [0.58–0.66] in Table 1 but [0.69–0.77] elsewhere, and ResNet3D is reported as [0.69–0.77] in Table 1 but [0.71–0.78] in the abstract and paragraph text. These are not minor editorial issues because they affect the paper’s main quantitative claim.

4) Verdict

The paper presents a clear applied case study and a potentially useful practical observation, namely that generic image-style augmentation may be poorly matched to violent-behavior recognition in surveillance video. However, the current version does not yet support a strong research contribution. The methodological novelty is limited, the main augmentation claim overlaps with already known concerns about semantic validity of video augmentation, the empirical evaluation is not positioned against current strong baselines, and the cross-dataset comparison does not fully isolate the source of the reported gains. The paper would be stronger with a clearer novelty claim, stronger comparison to recent public results, a controlled per-augmentation ablation, and corrected quantitative reporting.

---

### Decision · Program_Chairs · 2026-03-14

**Decision:**

Accept (Oral)

**Comment:**

Dear Author(s),

On behalf of the Program Committee of the International Conference on Mathematics of Artificial Intelligence (MathAI 2026), we are pleased to inform you that your paper has been accepted for an oral presentation at MathAI 2026.

Your paper was evaluated through a rigorous two-stage review process involving both automated screening and expert review by members of the Program Committee. The reviewers recognized the quality and contribution of your work.

Presentation details:

- Format: Oral presentation (15–20 minutes + 5 minutes Q&A)
- Mode: You may present either in person (offline) at the conference venue in Sirius, Russia, or remotely via Zoom. Please indicate your preferred mode when confirming your participation.
- Conference dates: Marh 30 - April 3, 2026
- Website: https://mathai.club

Next steps:

1. Please confirm your participation and presentation mode by replying to this email mathai.club@yandex.ru no later than March 15, 2026 18:00 Moscow time.
2. If you plan to attend in person, the organizing committee will provide accommodation details separately.
3. Please prepare your final camera-ready manuscript according to the formatting guidelines available at https://mathai.club and upload it to OpenReview by March 15, 2026 18:00 Moscow time.

Should you have any questions regarding the program, logistics, or your presentation slot, please do not hesitate to contact us.

We look forward to your contribution to MathAI 2026.

With kind regards,

MathAI 2026 Program Committee
International Conference on Mathematics of Artificial Intelligence
https://mathai.club
OpenReview: https://openreview.net/group?id=mathai.club/MathAI/2026/Conference
Telegram: https://t.me/MathAI_club
Email: mathai.club@yandex.ru